# The Release of Inflammatory Mediators from Acid-Stimulated Mesenchymal Stromal Cells Favours Tumour Invasiveness and Metastasis in Osteosarcoma

**DOI:** 10.3390/cancers13225855

**Published:** 2021-11-22

**Authors:** Sofia Avnet, Silvia Lemma, Margherita Cortini, Gemma Di Pompo, Francesca Perut, Maria Veronica Lipreri, Laura Roncuzzi, Marta Columbaro, Costantino Errani, Alessandra Longhi, Nicoletta Zini, Dominique Heymann, Massimo Dominici, Giulia Grisendi, Giulia Golinelli, Lorena Consolino, Dario Livio Longo, Cristina Nanni, Alberto Righi, Nicola Baldini

**Affiliations:** 1Department of Biomedical and Neuromotor Sciences, University of Bologna, 40127 Bologna, Italy; sofia.avnet3@unibo.it (S.A.); silvia.lemma3@unibo.it (S.L.); maria.lipreri2@unibo.it (M.V.L.); 2BST Biomedical Science and Technologies Lab, IRCCS Istituto Ortopedico Rizzoli, 40136 Bologna, Italy; margherita.cortini@ior.it (M.C.); gemma.dipompo@ior.it (G.D.P.); francesca.perut@ior.it (F.P.); laura.roncuzzi@ior.it (L.R.); marta.columbaro@ior.it (M.C.); 3Oncologic Orthopaedic Unit, IRCCS Istituto Ortopedico Rizzoli, 40126 Bologna, Italy; costantino.errani@ior.it; 4Osteoncology, Bone and Soft Tissue Sarcomas and Innovative Therapies, IRCCS Istituto Ortopedico Rizzoli, 40136 Bologna, Italy; alessandra.longhi@ior.it; 5CNR Institute of Molecular Genetics “Luigi Luca Cavalli-Sforza”, Unit of Bologna, via di Barbiano 1/10, 40136 Bologna, Italy; nzini@area.bo.cnr.it; 6IRCCS Istituto Ortopedico Rizzoli, 40136 Bologna, Italy; 7Institut de Cancérologie de l’Ouest, Tumor Heterogeneity and Precision Medicine, Université de Nantes, 44805 Saint-Herblain, France; Dominique.Heymann@univ-nantes.fr; 8Division of Oncology, Department of Medical and Surgical Sciences, University of Modena and Reggio Emilia, 41124 Modena, Italy; massimo.dominici@unimore.it (M.D.); giulia.grisendi@unimore.it (G.G.); giulia.golinelli@unimore.it (G.G.); 9Department of Molecular Biotechnology and Health Sciences, University of Turin, 10126 Torino, Italy; lconsolino@ukaachen.de; 10Institute of Biostructures and Bioimaging (IBB), Italian National Research Council (CNR), 10125 Torino, Italy; dariolivio.longo@cnr.it; 11Nuclear Medicine Unit, IRCCS Azienda Ospedaliero-Universitaria di Bologna, 40138 Bologna, Italy; cristina.nanni@aosp.bo.it; 12Surgical Pathology Unit, IRCCS Istituto Ortopedico Rizzoli, 40136 Bologna, Italy; alberto.righi@ior.it

**Keywords:** osteosarcoma, metastasis, acid microenvironment, mesenchymal stromal cells, inflammatory cytokines

## Abstract

**Simple Summary:**

We aimed to validate the correlation between tumour glycolysis/acidosis and inflammation in osteosarcoma-associated mesenchymal stromal cells and investigate the role of acidity-induced inflammation in the development of metastasis in this very aggressive cancer. We confirmed the presence of an acidic microenvironment in osteosarcoma xenografts, both subcutaneous and orthotopic, using state-of-the-art imaging technologies; corroborated the correlation between tumour glycolysis, acidosis, and inflammatory markers in human patients; and finally, explored the use of anti-IL6 antibody to target these pathogenic pathways, using advanced 3D microfluidic models. In the future, advanced imaging systems for the measurement of tumour glycolysis and/or pH may help identify osteosarcoma patients who would benefit from anti-IL6 therapies to complement conventional therapy.

**Abstract:**

Osteosarcoma is the most frequent primary malignant bone tumour with an impressive tendency to metastasise. Highly proliferative tumour cells release a remarkable amount of protons into the extracellular space that activates the NF-kB inflammatory pathway in adjacent stromal cells. In this study, we further validated the correlation between tumour glycolysis/acidosis and its role in metastases. In patients, at diagnosis, we found high circulating levels of inflammatory mediators (IL6, IL8 and miR-136-5p-containing extracellular vesicles). IL6 serum levels significantly correlated with disease-free survival and ^18^F-FDG PET/CT uptake, an indirect measurement of tumour glycolysis and, hence, of acidosis. In vivo subcutaneous and orthotopic models, co-injected with mesenchymal stromal (MSC) and osteosarcoma cells, formed an acidic tumour microenvironment (mean pH 6.86, as assessed by in vivo MRI-CEST pH imaging). In these xenografts, we enlightened the expression of both IL6 and the NF-kB complex subunit in stromal cells infiltrating the tumour acidic area. The co-injection with MSC also significantly increased lung metastases. Finally, by using 3D microfluidic models, we directly showed the promotion of osteosarcoma invasiveness by acidosis via IL6 and MSC. In conclusion, osteosarcoma-associated MSC react to intratumoural acidosis by triggering an inflammatory response that, in turn, promotes tumour invasiveness at the primary site toward metastasis development.

## 1. Introduction

Osteosarcoma (OS) is a highly malignant bone tumour of mesenchymal origin that primarily affects children and adolescents, with an impressive tendency to metastasise. OS is resistant to existing treatments in at least 40% of patients. Over the past 40 years, despite significant efforts, no improvement in prognosis has been achieved [1,2,3]. The lack of appropriate preclinical models, the complexity of OS biology, and an insufficient consideration of the multiple factors that modulate the behaviour of OS in its microenvironment are possible causes that should be addressed and overcome.

Changes in physical, chemical, and biochemical parameters are crucial determinants of tumour microenvironment (TME) [4]. Among them, hypoxia [5] and acidosis are recognised hallmarks of cancer and significantly influence tumour cell behaviour and clinical outcome [6]. Interstitial acidification results from the combination of dysfunctional tissue perfusion and from the increased glycolytic activity of highly proliferating cancer cells [6,7], leading to the accumulation of extracellular protons. Dysregulation of tumour extracellular pH is a common feature of several cancers [8]. In breast carcinoma, intratumoural acidosis and upregulation of glucose metabolism was directly correlated with in vivo imaging, through the combination of positron emission tomography (PET) and magnetic resonance (MR)-chemical exchange saturation transfer (CEST) pH imaging [9]. We demonstrated that sarcomas have a high glycolytic activity [10], which in turn induces the expression of several ion/proton pumps and transporters that strongly acidify the extracellular space [11], and causes profound metabolic adaptation [12,13]. Furthermore, we have found mRNA co-expression of indirect markers of glycolysis and acidosis, the glucose transporter 1 (GLUT1) and the V0c subunit of the vacuolar ATPase (V-ATPase), respectively [10]. Finally, in TME of OS, a low pH may also be promoted by osteoclasts (OC) during tumour-induced bone resorption [14].

Tumours are heterogeneous lesions whose growth and progression largely depend on reciprocal interactions between genetically altered neoplastic cells and their non-neoplastic microenvironment [15], and normal cells may be altered by tumour-derived modifications of the extracellular microenvironment, such as extracellular acidosis. Hence, acid-derived effects may be mediated by the reaction of tumour-associated stroma. Among the reactive components, endothelial, immune, and mesenchymal stromal cells (MSC)/cancer-associated fibroblasts (CAF) are frequently observed [15,16,17,18,19] and are crucial for the progression of the disease [20,21]. Notably, OS develops within bone, a tissue with a high concentration of mesenchymal progenitors. Circulating MSC from the bloodstream are strongly attracted at the site of OS development, possibly contributing to a rapid tumour expansion [20]. The interaction between OS cells and the reactive elements induces the secretion of soluble factors such as cytokines, growth factors, micro- or nanovesicles [22], metabolites [23], and matrix components, which further define OS TME and progression. In this context, it is important to emphasise that acidosis is a stress and pro-inflammatory signal for stromal cells, even in the absence of malignancy [24]. Indeed, a low pH is associated with inflammation [25,26,27] and is a stimulus for tissue healing. By analogy, cancer is considered as a wound that never heals [28]: similar to what happens in hypoxic regions, acidosis chronically stimulates the nuclear factor kappa-light-chain-enhancer of activated B cells (NF-kB)-mediated inflammatory response of the tumour-associated mesenchymal stroma. This, in turn, leads to the secretion of cytokines, chemokines, and growth factors that may further elicit cancer invasiveness and stemness, thereby resulting in a high metastatic incidence [24,29]. Although intriguing, this hypothesis has never been demonstrated through in vivo preclinical studies or using 3D models to adequately recreate the TME. 

In this study, we confirm the correlation between the acid-induced release of inflammatory mediators and the tumour aggressiveness/glycolytic activity in OS patients. Then, we demonstrate the formation of an acidic TME in an animal model that recapitulates the TME of OS, including the infiltrating MSC at the primary site, the activation of inflammatory pathways, and the occurrence of spontaneous metastases. Finally, the results obtained in vivo are further validated in vitro using mixed cultures of 3D microfluidic MSC/tumour cells.

## 2. Materials and Methods

### 2.1. Observational Clinical Pilot Study

Human blood was collected from OS patients in a continuous and homogeneous series, from 2015 to 2018. Blood samples were anonymised during processing. Blood was collected into SST™ tubes (#368968 BD Vacutainer™, ThermoFisher, Waltham, MA, USA) and serum was obtained by low-speed centrifugation (750*× g* for 8 min) and stored at −80 °C prior to analysis. The inclusion criteria were a diagnosis of a histologically confirmed OS. Exclusion criteria were the presence of metastasis at diagnosis, as ascertained by X-ray analysis. Serum was obtained at the time of diagnosis (T0). Patients were then assigned to chemotherapy and serum was collected again at the end of treatment (T1), in a disease-free setting. T1 was, in most cases, collected around 12 months after T0. All clinical and pathological data were recovered and examined (Table 1).

### 2.2. Circulating Markers and ELISA of IL6 and IL8

C-reactive protein (CRP) serum concentrations were retrieved from medical records. The levels of interleukin 6 (IL6) and 8 (IL8) in the serum of patients were measured by human IL6 and human CXCL8/IL8 Quantikine ELISA kits (RD System, sensitivity: 0.7 pg/mL and 7.5 pg/mL, respectively). Obtained values were correlated with (18)F-fluorodeoxyglucose positron emission tomography/computed tomography (^18^F-FDG PET/CT) analysis, when available. Whole-body ^18^F-FDG PET/CT scans were carried out following standard procedures. Following a 6 h fast, 3 MBq/kg of ^18^F-FDG was intravenously injected in patients. The uptake time was 60 ± 10 min in all patients on a 3D tomography (Discovery STE; GE or Discovery MY) for 2 min per bed position. Cross-calibration was performed using an image quality NEMA phantom. A low-dose CT scan (100 kV, 120 mA) was performed both for attenuation correction and as an anatomical map. For each scan, metabolic tumour volume (MTV) was measured. MTV measurement was calculated on PET/CT images using a semi-quantitative analysis (40% threshold).

### 2.3. 2D Cell Cultures 

143B human OS cell line was purchased from the American Type Culture Collection (ATCC). For transfection with the luciferase reporter gene, cells were seeded at 10,000/cm^2^ into a 48 multiwell plate and grown in complete medium for 24 h. Then, culture medium was replaced with fresh complete medium containing 20 MOI of RedFect Red-Fluc-Puro Lentiviral Particles (PerkinElmer, Waltham, MA, USA) and 4 mg/mL of polybrene (Sigma-Aldrich, Saint Luis, MO, USA). Twenty-four hours after infection, supernatant containing viral particles was discarded and replaced with fresh culture medium. After infection, tumour cells were treated with 1 μg/mL of puromycin (Sigma) for 1 week in order to select the tumour cells stably expressing Luciola Italica Luciferase gene. Adipose-derived MSC (AD-MSC) were obtained as previously described from lipoaspirate specimens of individuals undergoing liposuction for aesthetic purposes [30]. To obtain green fluorescent protein (GFP)-AD-MSC, AD-MSC were transduced by virus-containing medium from FLYRD18-GFP packaging cell line (~1 × 10^6^ transducing units/mL). Supernatant collected from FLYRD18-GFP and containing viral particles was diluted at a 1:1 ratio with fresh culture medium (DMEM, 10% foetal calf serum 1% penicillin and streptomycin, 1% glutamine) and added with polybrene (6 μg/mL). AD-MSC were incubated with virus-containing medium for 6 h. Infection was repeated for 3 consecutive days [30,31]. Cells were cultured in RPMI plus 20 units/mL penicillin, 100 μg/mL streptomycin, and 10% foetal calf serum (FCS) at pH 7.4 (complete medium), and incubated at 37 °C in a humidified 5% CO_2_ atmosphere. When different pH values were used, cells were seeded in complete medium, and after 24 h, the medium was changed. To pre-set the pH value in 5% CO_2_ atmosphere, we at the mediums different concentrations of sodium bicarbonate, according to the Henderson–Hasselbach equation. At the endpoint of each experiment, the final pH in the supernatant was always measured by a digital pH-meter (pH 301, HANNA Instruments, Woonsocket, RI, USA).

### 2.4. Spheroids

To obtain 3D spheroids, we used the hanging-drop method, as previously described [12]. Briefly, luc-143B cells were obtained from 70% confluent flasks, detached and counted. Cells were plated on a 96-well Round Bottom Ultra-low attachment plate (Costar Corning, Corning, NY, USA) in 200 µL of filtered pH 7.4 or unbuffered RPMI. An additional 200 µL of medium was added to each well and the lid was placed over the plate, with specific supports fixed at every corner. The plate was flipped and incubated in gentle shaking conditions at 37 °C and 5% CO_2_ overnight. The following day, the plate was flipped again, 200 µL of medium were removed and the time point was indicated as T0; the spheroids were then allowed to grow for an additional 96 h. Homotypic luc-143B spheroids were obtained seeding 5 × 10^3^ cells/well, whereas heterotypic luc-143B/GFP-AD-MSC were seeded in a 1:3 ratio (5 × 10^3^ tumour cells + 1.5 × 10^4^ stromal cells). We used such a high ratio of MSC to tumour cells because 143b cells have a much higher proliferation rate and, within days, this ratio likely reverses completely. For 3D models, to generate neutral-grown spheroids, RPMI was added with sodium bicarbonate and adjusted to pH 7.4, whereas, to obtain acidic-grown spheroids, the medium was left unbuffered without the supplementation of sodium bicarbonate. This allowed the spheroids to adjust the pH according to their density and growth. For all the experiments performed in 3D, spheroids were thus grown for 96 h at pH 7.4 or unbuffered medium.

### 2.5. Immunofluorescence 

For immunofluorescence staining, spheroids were grown for 5 days, then were collected and cytocentrifuged at 800 rpm for 5 min on glass coverslips and next fixed in ice-cold methanol for 20 min. Cells were incubated with anti-connexin-43 antibody (1:500, Abcam, Cambridge, UK) overnight at 4 °C, followed by a secondary anti-mouse TRITC-conjugated antibody (1:500, Chemicon, Limburg an der Lahn, Germany). Nuclei were counterstained with Hoechst 33342. Images were acquired with an objective 20x air, numerical aperture 0.75, Galvano scanning, zoom at 1, line average of 4 (A1R MP confocal microscope, Nikon, Tokyo, Japan, scale bar 50 μM).

### 2.6. 3D Cell Cultures: Microfluidic Models

(a)Single-cell invasion ability: A single-cell suspension of luc-143B cells (1 × 10^3^ cells/channel) was obtained from the 70% confluent flask, detached in trypsin and counted. Cells were stained with Vibrant DIL Cell-Label (Life Technologies, Carlsbad, CA, USA) for 20 min at 37 °C following the manufacturer’s procedure. Cells were then mixed in a 1:3 ratio with GFP-AD-MSC and resuspended in 50% Matrigel^®^, 25% neutralised Rat Tail Collagen I (Cultrex, RD Systems), and 25% cell culture medium supplemented with 0.1% FCS. Cells embedded in the gel were added in the lower channel of a microfluidic 2-lane OrganoPlate^®^ (Mimetas, Oegstgeest, The Netherlands) and placed for 15 min a 37 °C to allow for gel polymerisation. The channel was then filled with culture medium at neutral (7.4) or acidic (6.0) pH. As a stimulus for cell migration, the upper perfusion channel was added with 10% FCS medium, and Recombinant Human IL6 (Peprotech, 50 µg/mL) was used as a positive control. To simulate physiological perfusion flow, the OrganoPlate^®^ was placed on a rocking plate: the rocker was set to change the 7° tilting angle every 8 min. Cell migration was visualised with confocal microscopy after 24 h. Only DIL-positive cells migrated in the perfusion channel were counted. Images were acquired with an objective 20x air, numerical aperture 0.75, Galvano scanning, zoom at 1, line average of 4, A1R MP confocal microscope, Nikon, scale bar 50 μM. The total acquired Z-stack was 161 µm, with a Z-step of 1.67 µm: 2 images/channel on 5 different channels.(b)Cellular escape from spheroids: We obtained heterotypic spheroids (luc-143B/GFP-AD-MSC) with the hanging-drop method after 96 h of culture in RPMI pH 7.4, as already described. The spheroids were placed in the open grafting chamber of a microfluidic OrganoPlate^®^ Graft (Mimetas,), previously filled with 50% Matrigel^®^, 25% neutralised Rat Tail Collagen I (Cultrex, RD Systems), and 25% cell culture medium supplemented with 0.1% FCS. The chamber was then covered by 0.1% FCS medium at neutral (7.4) or acidic (6.0) pH. The lateral channels were filled with 10% medium to provide a stimulus for cell invasion. Cells were allowed to invade the gel for 7 days, with or without the anti-IL6 antibody (Tocilizumab, 100 µg/mL, Roche) that was added daily to the grafting chamber. At the endpoint, cells were fixed with 3.7% paraformaldehyde and nuclei were stained with Hoechst 33342 (Sigma). Images were acquired with an objective 20x air, numerical aperture 0.75, Resonant scanning, zoom at 1, line average of 4 (A1R MP confocal microscope, Nikon, scale bar 50 μM). The total acquired Z-stack was 200 µm, with a Z-step of 5 µm: 4 images/chamber were acquired and only cells visualised inside the gel were counted. The experiment was performed with three replicates.

### 2.7. Extracellular Nanovesicle Isolation 

(a)Isolation of extracellular nanovesicles (EV) from AD-MSC: We assessed cell density and viability of AD-MSC at different pH (6.8 or 7.4) by the erythrosine B (Sigma-Aldrich) dye exclusion method. AD-MSC were grown in complete medium until 70–80% confluence. Cells were washed with Phosphate-Buffered Saline (PBS) and then incubated for 48 h with serum-free α-MEM at pH 6.8 or 7.4 (Sigma-Aldrich). The supernatant was collected after two consecutive periods (18 h and an additional 18 h) from AD-MSC grown on 20 Petri dish (diameter 150 mm, 18 mL/Petri). The EV were purified by differential centrifugation: 500× *g* for 10 min (twice), 2000× *g* for 15 min (twice), and 10,000× *g* for 30 min (twice) at 4 °C to remove floating cells and cellular debris. The supernatant was then ultracentrifuged at 110,000× *g* for 1 h at 4 °C. The EV pellet was resuspended in PBS and centrifuged at 110,000× *g* for 1 h at 4 °C (Beckman Coulter, Milan, Italy). The EV pellet was resuspended in PBS and stored at −80 °C until use. EV quantity was determined by the Bradford method (Bio-Rad, Milan, Italy).(b)Isolation of EV from serum (see Table 1): EV were precipitated from the serum (600 µL) by using the miRCURY Exosome Isolation Kit—Serum and plasma (QIAGEN, Hilden, Germany)

### 2.8. RNA Isolation and qPCR 

(a)Extracellular nanovesicles: miRNA isolation from EV isolated from serum and their analysis were conducted by QIAGEN Genomic Services. Total RNA was isolated from serum-derived EV by using the exoRNeasy Serum/Plasma Kit (QIAGEN) according to the manufacturer’s instructions. Total RNA (7 µL) was reverse transcribed in 35 µL reactions using the miRCURY LNA RT Kit (QIAGEN). cDNA was diluted 50× and assayed in 10 µL PCR reactions following the protocol for miRCURY LNA miRNA PCR; each miRNA was assayed once by qPCR on the miRNA Ready-to-Use PCR, Serum/Plasma Focus panel (192 miRNA assay) by using the miRCURY LNA SYBR Green master mix. Negative controls excluding the template from the reverse transcription reaction were performed and profiled like the samples. The amplification was performed in a LightCycler^®^ 480 Real-Time PCR System (Roche) in 384-well plates. The amplification curves were analysed using Roche LC software, both for determination of Cq (by the 2nd derivative method) and for melting curve analysis. All assays were inspected for distinct melting curves and the Tm was checked to be within known specifications for the assay. Furthermore, assays must be detected with 5 Cq less than the negative control, and with Cq < 37 to be included in the data analysis. Data that did not pass these criteria were omitted from any further analysis. Cq was calculated as the 2nd derivative. All data were normalised to the average of assays detected in all samples [32]. Total RNA was isolated also from AD-MSC-derived EV by using miRCURY RNA Isolation Kits (QIAGEN) and reverse transcribed by using the miRCURY LNA RT Kit (QIAGEN). cDNA was diluted 20× and assayed in 10 µL PCR reactions according to the protocol for miRCURY LNA miRNA PCR, by using the hsa-miR-136-5p miRCURY LNA miRNA PCR Assay and miRCURY LNA SYBR Green master mix (QIAGEN). RNA spike-in templates (UniSp2, UniSp4 and UniSp5) and synthetic transcript UniSp6 (QIAGEN) were used as controls. The amplification was performed in a Bio-Rad CFX96 Touch System (Bio-rad, Hercules, CA, USA) in 96-well plates.(b)From tumour tissues and cell cultures: Total RNA was collected from frozen xenografts that were powdered by using a Mikro-Dismembrator (B. Braun Biotech International, Melsungen, Germany) with TRIzol reagent (ThermoFisher Scientific, Waltham, MA, USA). Total RNA was reverse transcribed into cDNA using RNase inhibitor and MuLV Reverse Transcriptase (Applied Biosystems, Foster City, CA). First-strand cDNA was synthesised by RT-qPCR using random hexamers. Real-time Polymerisation Chain Reaction (Real-time PCR) was performed by amplifying 500 ng of cDNA using the Light Cycler instrument and the Universal Probe Library system (Light Cycler instrument, Roche Applied Science, Monza, Italy). Probes and primers were selected by using a web-based assay design software (ProbeFinder, https://www.roche-applied-science.com, accessed on 18 October 2021). All primers were validated by BLAST (National Centre for Biotechnology Information) and Oligo Primer Analysis Software (Oligo, Colorado Springs, CO, USA); for primer information, see Appendix A. The amplification protocol was 95 °C for 10 min, 45 cycles (95 °C for 10 s, 60 °C for 30 s, and 72 °C for 1 s); 40 °C for 30 s. Relative gene expression was obtained with the ratio with the geometric mean of the different housekeeping genes (HKG) (Gusb, YWHZ, 18S RNA, GAPDH) according to the ΔΔCt model, as previously described [24]. These housekeeping genes were considered suitable for the evaluation of mRNA expression under acidic conditions, as previously demonstrated [33]. All the experiments were replicated three times.

### 2.9. Electron Microscopy 

(a)Analysis of EV: Transmission electron microscopy was used to evaluate the morphology and size of EV, isolated from both human serum and cell cultures. Briefly, EV were resuspended in 2% paraformaldehyde and deposited onto formvar-carbon-coated grids. Then, EV were fixed in 1% glutaraldehyde, washed, counterstained with a solution of uranyl oxalate, pH 7.0, and embedded in a mixture of 4% uranyl acetate and 2% methylcellulose before observation with a Zeiss-EM 109 transmission electron microscope (Zeiss, Jena, Germany). Minor modification of this protocol was applied for electron microscopy analysis of serum-derived EV, as previously described [34]; the diameter of EVs was measured, and the percentage of size distribution was calculated.(b)Analysis of spheroids: Spheroids, obtained as described above, were fixed with 2.5% glutaraldehyde in 0.1 M cacodylate buffer pH 7.6 for 1 h at room temperature, post-fixed with 1% OsO_4_ in cacodylate buffer for 1 h, dehydrated in an ethanol series and embedded in Epon resin. Ultrathin sections stained with uranyl acetate and lead citrate were observed with a Jem-1011 transmission electron microscope (Jeol Inc., Peabody, MA, USA).

### 2.10. Western Blotting

EV and cell pellets were lysed with RIPA lysis buffer (25 mM Tris-HCl pH 7.6, 150 mM NaCl, 1% NP-40, 1% Na-deoxycholate, 0.1% SDS) and protease inhibitor cocktail (Roche, Milan, Italy) for 30 min at 4 °C. Cell debris and nuclei were removed by centrifugation. The protein concentration was determined using the Bradford assay (Bio-Rad). The total cellular and EV proteins were resolved by 10% SDS-polyacrylamide gel and transferred to a nitrocellulose membrane (Thermo Fisher Scientific). After blocking with 5% dry milk (Thermo Fisher Scientific) in T-TBS (0.1 M Tris-HCl pH 8.0, 1.5 M NaCl and 1% Tween-20) for 1 h at room temperature, the membranes were incubated with CD63 and hsp70 (1: 1000) rabbit polyclonal antibodies (System Biosciences, #EXOABKIT1) overnight at 4 °C. After vigorous washing in 0.05% Tween-20 in PBS, the membranes were incubated for 1 hr at R°T with goat anti-rabbit antibody (System Biosciences, #EXOABKIT1) (1:20,000) conjugated to horseradish peroxidase that was diluted in 5% dry milk in T-TBS. The ECL Western blot analysis system (Euroclone) was used to detect the immunocomplexes.

### 2.11. In Vivo Models

For the in vivo models, we used both subcutaneous and intratibial injections of homotypic cell population or of heterotypic tumour/stromal mixed cell populations (Table 2). As for the spheroids, we used such a high ratio of MSC to tumour cells because 143b cells have a much higher proliferation rate and, within days, this ratio likely reverses completely.

Mice were anesthetised with isoflurane vaporised with O_2_. Isoflurane was used at 3.0% for induction and at 1.0% to 2.0% for maintenance. All the procedures involving the animals were conducted according to national and international laws on experimental animals (L.D. 26/2014; Directive 2010/63/EU) and the approved experimental protocol procedure.

(a)Subcutaneous model of OS: NOD/SCID animals were housed and maintained in a pathogen-free environment. The 5-week-old male NOD/SCID mice (Charles River Laboratories International, Wilmington, MA, USA) were randomly split into two groups for the subcutaneous injection in the flank of homotypic or heterotypic cell populations, mixed with reduced growth factor Matrigel^®^ (BD Life Sciences, Biosciences, Franklin Lakes, NJ, USA). For the number and type of cells, see Table 1 (experiment 1 and 2). For group #2, an isotonic solution containing GFP-AD-MSC was injected in the tail vein, on day 7 after surgery, once a week. Weights were taken daily during treatment. Mice were euthanised when tumour volume exceeded 2500 mm^3^.(b)Orthotopic model of OS: NOD/SCID animals were housed and maintained in a pathogen-free environment. The 5-week-old male NOD/SCID mice (Charles River Laboratories International) were randomly split into two groups and injected with luc-143B cells, w/o GFP-AD-MSC (see Exp. 4 in Table 1), in the left tibia. Cells were suspended in an isotonic saline solution. As previously described [22], the skin at the tibial plateau of the left calf was cut and a sterile incision was carefully made to expose the tibia. Next, 10 µL of cell suspension was slowly injected into the medullary cavity of the left tibia. The micro-syringe was then removed, and bone wax was used to seal the hole. Finally, the aseptic incision was sutured with vicryl polyglactin. Mice were sacrificed at different time points (*n* = 1/group at 8 days only for group #4; *n* = 4/group at days 12 days, *n* = 4/group at 15 days, *n* = 4/group at 18 days). In any case, mice were euthanised if the tumour volume exceeded 2500 mm^3^ or when mice showed signs of lung metastases development (respiratory distress, weakness, weight loss, dorsal kyphosis) or were positive for the luciferase signal, as described in the following section. The tibia and the lungs were extracted for histological and macroscopic analysis.

### 2.12. In Vivo MRI-CEST Intratumoural pH Imaging

MR images were acquired on a 7T Bruker MicroImaging (Bruker, Ettlingen, Germany) scanner, equipped with a 30 mm 1 H quadrature RF coil. During all the experiments, a mixture of xylazine 5 mg/kg (Rompun, Bayer, Milan, Italy) and tiletamine/zolazepam 20 mg/kg (Zoletil 100, Virbac, Milan, Italy) was administered via intramuscular injection to anesthetise mice. We maintained a constant temperature and monitored the breath rate by an air pillow placed below the animal (SA Instruments, Stony Brook, NY, USA). A single T2-weighted axial slice crossing the centre of the tumours was acquired with TR 4 s, TE 3.7 ms, NA 1, slice thickness 1.5 mm, FOV 30 × 30 mm, matrix size 256 × 256, which yielded an in-plane resolution of 117 μm. Z-spectra were acquired using a single-shot RARE sequence with centric encoding (typical setting TR/TE/NEX 6.0 s/4.14 ms/1) that was preceded by a 3 µT cw block presaturation pulse, for 5 s, and by a fat-suppression module. A series of 40 MR frequencies were saturated to acquire a CEST spectrum in the frequency offset range ±10 ppm. We used an acquisition matrix of 96 × 96 reconstructed to 128 × 128 for a field of view of 3 × 3 cm^2^ (in-plane spatial resolution = 234 µm, slice thickness = 1.5 mm). MRI-CEST pH mapping was performed by acquiring Z-spectra before and after intravenous injection of iopamidol (4 g iodine/kg body weight, kindly provided by Bracco Imaging SpA, Colleretto Giacosa, Italy) into the tail vein through a 27-gauge needle catheter. All CEST images were analysed using a homemade script implemented in MATLAB (The Mathworks, Inc., Natick, MA, USA). The Z-spectra were interpolated, on a voxel-by-voxel basis, by smoothing splines, B0-shift corrected; saturation transfer efficiency (ST%) was measured by punctual analysis at 4.2 and at 5.5 ppm [35]. Difference contrast maps (ΔST%) were calculated at 4.2 and at 5.5 ppm by subtracting the ST% contrast after iodinated contrast medium injection from the ST contrast before the injection on a per voxel basis in order to reduce the confounding effect of the endogenous contributions. We set up a threshold value of 1% to discriminate between enhancing and not enhancing pixels. We estimated extracellular tumour pH values in vivo by applying a ratiometric procedure. Tumour pH maps were then superimposed onto the anatomical reference image [36].

### 2.13. H&E and Immunostaining

Tumour xenografts or lungs were fixed in 4% formaldehyde and embedded in paraffin. Only for tibia in orthotopic mice, samples were also decalcified by using an EDTA solution (0.5 M pH 8). For all types of samples, 5 μm sections were mounted on a glass slide covered with 2% silane solution in acetone, and stained with haematoxylin/eosin. In contrast, for immunostaining, after dewaxing in Citro Histoclear (Histo-Line Laboratories, Milano, Italy) and rehydration in ethanol, tissue sections were incubated in a 3% hydrogen peroxide solution, followed by an incubation in a 2% bovine serum albumin solution in order to block the endogenous peroxidases and nonspecific binding. We then incubated the slides with a rabbit anti-vimentin (1:100, sc-6260, Santa Cruz Biotechnology, Inc. Dallas, TX, USA), rabbit anti-ATP6V1B2 (1:500, HPA008147, Sigma) or rabbit anti-RelB (1:250, HPA040506, Sigma) primary antibody. Subsequently, the slides were incubated with a biotinylated secondary antibody, covered with DAB, and counterstained with Mayer’s haematoxylin (EnVision FLEX, High pH Link visualisation system, Agilent Technologies, Santa Clara, CA, USA). 

### 2.14. Quantitation of Tumour Growth and Metastasis

(a)Tumour growth at the site of injection: Tumour growth was assessed at 8 days, using the bioluminescence generated by the injected luciferase-expressing luc-143B cells in mice (*n* = 6) as the read-out. At the endpoint, mice were injected intraperitoneally (150 μg/g of body weight) with a D-luciferin (Promega, Madison, WI, USA) solution in PBS, anaesthetised with isofluorane and imaged using an IVIS in vivo imaging system (PerkinElmer) with Living Image software (version 4.3.1, Perkin Elmer). Images were taken within the first 12 min after D-luciferin administration.(b)Metastatic formation: We used both luciferase assay in live animals, as described above, and vimentin immunostaining analysis to enlighten the development of tumour clones in the lung (using tissue sectioning followed by standard light microscopy techniques) after the mice’s sacrifice. Results that could not be confirmed with both immunohistochemistry and bioluminescence were excluded from the analysis. As parameters for the evaluation of metastasis, we used the tumour area (as measured by the quantification of the area of the lung that was positive to vimentin signal with respect to the total lung area), the number of tumour lesions (as measured by the quantification of the number of spots associated with the vimentin signal), and survival. Tumour area and the number of tumour lesions were analysed from three distinguished lung areas (top, middle and bottom). For survival analysis, uncensored corresponded to mice that were positive for both luciferase signal and vimentin (*n* = 8 mice for group #3 and *n* = 9 mice for group #4). The remaining mice were excluded.

### 2.15. Statistics

Because of the small number of observations, data were not considered as normally distributed and non-parametric tests were used. By using GraphPad Prism 7 software, we analysed the difference between two groups by the Mann–Whitney U test for unpaired analysis, or the Wilcoxon test for paired analysis, and the correlations by the Spearman Rank test. Outliers assessed by the Grubbs test (alpha = 0.05) were excluded from the analysis. The normal distribution of miRNA content in serum was evaluated by the Shapiro–Wilk normality test, and the differential expression between two groups of miRNAs detected by a paired *t*-test (with pairing factor Pair). Both raw *p*-values and *p*-values adjusted for multiple testing by the Benjamini–Hochberg correction are reported. For the survival analysis (disease recurrence) in the observational human study, we used the Kaplan–Meier curve, Log-rank Mantel–Cox test. High IL6 was considered ≥4.3 pg/mL (50th percentile of the total patient group). CRP value was considered >2 mg/dL (90th percentile of the total patient group). The average follow-up time was 27.66 ± 3.16 for the high-IL6 group, 21.08 ± 3.79 for the high-CRP group, and 26.36 ± 1.79 for the others. The Kaplan–Meier curve, Log-rank Mantel–Cox test was also used for the evaluation of survival in the animal experiment (lung metastasis formation). Data were expressed as the means ± standard error (SEM) unless otherwise specified, and only *p*-values < 0.05 were considered as statistically significant.

## 3. Results

### 3.1. Acid-Related Systemic Inflammatory Mediators in Osteosarcoma Patients

In this study, we hypothesised that MSC exposed to extracellular acidity at the tumour site may release—locally and, subsequently, systemically—high levels of inflammatory mediators that promote the onset of lung metastasis. We focused on two inflammatory cytokines, IL6 and IL8, which, according to our previous studies, are significantly secreted in acid-stimulated MSC with respect to neutral conditions [24]. By ELISA, we thus analysed the serum levels of IL6 and IL8 in a series of non-metastatic OS patients at diagnosis, before treatment (T0) and at the end of the therapeutic protocol (T1), after tumour removal (at 13.94 ± 0.76 months since T0), and found a significant decrease in both IL6 and IL8 at T1 (Figure 1A).

Although not statistically significant, we observed a trend of decrease for the percentage of disease-free survival in the high-IL6 group of patients with respect to the low-IL6 group (Figure 1B). Confirming earlier reports [37,38], in our current series of patients, we also found that both inflammatory markers C-reactive protein (CRP) and IL6 were associated with an increased risk of lung metastasis (Figure 1C, median survival for high IL6 is 31.47 months; for high CRP, 28.43 months), although this was not the case for IL8. In other cancer models, the rate of glucose uptake measured by ^18^F-FDG PET/CT analysis is directly related to the level of intratumoural acidosis. [9]. Thus, we used ^18^F-FDG PET/CT in the same patients as an indirect marker of high glycolysis and acidosis and indirectly demonstrated that in OS, circulating levels of the inflammatory cytokine IL6 also correlate with these two metabolic activities (Figure 1D). On the opposite side, we did not find a significant direct correlation with IL-8 serum levels. As a response to intratumoural acidosis, freely available cytokines may not be the only circulating inflammatory mediators secreted by tumour-infiltrating MSC that may be involved in the pathogenesis of OS. Most recently, MSC-derived EV containing miRNA have also been widely considered because they play an important role in cellular communication and macromolecule transmission [39]. We therefore looked for circulating EV, and we analysed their morphology and size (Figure 1E,F), and miRNA content. As with the assessment of inflammatory cytokines, we compared the expression level of the respective miRNA at T1 vs. T0 to highlighting those miRNAs that may have been derived directly from tumour-related events, including intratumoural acidosis. We detected 120 microRNAs and, among them, for 20 miRNA, T-paired analysis showed a significant difference between the two endpoints (Table 3). 

Moreover, the more stringent Benjamini–Hochberg (BH) correction test marked the most significant difference for miR-376a-3p, miR-154-5p, miR-136-5p and miR-376c-3p (Table 3). Notably, a pro-inflammatory activity for miR-136-5p has been reported: its overexpression, in rats, under non-cancer-related inflammatory conditions, caused the release of interleukin 1 beta (IL1β), IL6, tumour necrosis factor alpha (TNF-α), interferon alpha (IFNα), ikappaB-alpha kinase beta (IKKβ), via NF-κB activation [40,41]. Using normal AD-MSC in vitro as a model, we thus determined whether miR-136-5p found in patient serum could derive from acid-stimulated, tumour-associated stroma. For this purpose, we exposed AD-MSC to acidic conditions (pH 6.8). EV isolated from AD-MSC were similar in size to exosomes (30–100 nm) under both pH conditions (Figure 1E,F), and expressed the CD63 tetraspanin, a marker of EV (Figure 1G and Appendix A). The detected expression levels of CD63 were also significantly higher in EV preparations than in cell lysates. In contrast, cytosolic heat-shock protein 70 (hsp70), which is normally not expressed by EV, was only detected in cell lysates (Figure 1G and Appendix A). Furthermore, by miRNA qPCR analysis, we also confirmed the presence of miRNA 136-5p in ADMSC-derived EV at both pH values [42]. Finally, the release of EV by AD-MSC, and as a consequence, also of miRNA-136-5p was significantly higher at pH 6.8 compared to pH 7.4, as shown by EV total protein quantification in AD-MSC (Figure 1H). The increase in the secretion of EV under acidosis has been already reported [43]. 

In conclusion, in agreement with our previous results [24], we found increased levels of circulating inflammatory mediators in patients at the onset of OS, including the already documented inflammatory cytokines IL6 and IL8, and the lesser established pro-inflammatory miRNA 136-5p in circulating EV. According to our previous and current data on preclinical models, the release of these inflammatory mediators into the circulation may result from MSC infiltrating OS and stressed by intratumoural acidosis. 

### 3.2. Intratumoural Acidosis Induces IL6 and IL8 Expression in Tumour-Associated Mesenchymal Stroma in a Subcutaneous Model of OS

To confirm the hypothesis of the induction of an inflammatory phenotype in MSC by intratumoural acidosis in vivo, we then developed a subcutaneous model of OS (Table 2, trial n. 1, group #1; Figure 2A). First, we validated the model by assessing the average intratumoural pH after injection of the luc-143B tumour cells alone. Figure 2A displays representative images of the tumour extracellular pH map, as measured by MRI-CEST, of the induced tumour in each mouse. Although within the single tumour, pH was quite heterogeneous, the average intratumoural pH was highly reproducible (average pH 6.86 ± 0.05).

We then investigated how tumour-derived intratumoural acidosis can modulate the behaviour of MSC. We thus subcutaneously co-injected both GFP-AD-MSC and luc-143B cells and compared the results with mice injected only with tumour cells (Table 2, trial n. 2; Figure 2B). After H&E staining, group #2 (with GFP-AD-MSC) showed the presence of cells with a fibroblast-like morphology, forming septum-shaped structures (Figure 2B) that were absent in group #1 (w/o GFP-AD-MSC). As assessed by the quantification of the luciferase signal (tumour flux), group #2 also developed a significantly larger tumour (Appendix A). To dissect the role of intratumoural acidosis in mediating MSC-induced increased tumour growth, we in vitro cultured 3D spheroids under buffered neutral conditions or under unbuffered conditions, the latter intended to mimic the in vivo intratumoural acidosis. Indeed, as verified by using a microelectrode, when mixed luc-143B/GFP-AD-MSC spheroids were cultured in unbuffered medium, the pH of the supernatant spontaneously reached a value between 6.6 and 6.7. To test the proliferation rate of 143B cells, we directly counted tumour cells by identifying GFP-negative cells that, however, were positive for the proliferation marker Ki-67. As shown in Appendix A, GFP-AD-MSC strongly promoted luc-143B cell proliferation, but only when spheroids were cultured in the presence of neutral pH. 

Finally, in the animal models of trial n.2 (Figure 2B), we also evaluated the level of mRNA expression of IL6 in xenograft tissues and correlated them to the expression of the proton pump V-ATPase, isoform V1B2, as an indirect marker of acidosis [44]. Notably, although IL6 mRNA intratumoural levels were higher in group #1 (w/o AD-MSC) with respect to group #2 (with AD-MSC) (0.0111 ± 0.0025 vs. 0.0052 ± 0.0007, respectively, * *p* = 0.0259), it significantly correlated with the V1B2 expression only in group #2 (with AD-MSC, Figure 2C). 

In summary, our results suggest that the TME in the OS xenograft is acidic and that intratumoural acidosis induced the release of IL6 by tumour-associated MSC.

### 3.3. Reactive MSC in the TME Promote Lung Metastasis in an Orthotopic Model of OS

To validate the studied mechanism in a more realistic model of OS development and progression, we then moved to an orthotopic model of OS that spontaneously gives rise to lung metastases [22] (Figure 3A). First, we confirmed that the TME in this model reached the same intratumoural acidic pH of the subcutaneous model, by MRI-CEST pH imaging, with an average pH of 6.88 ± 0.04 (Figure 3B). According to the obtained results, by comparing the MRI-CEST pH imaging analysis of groups #1 and #3 (see Table 1), neither the site of injection nor the co-injection with GFP-AD-MSC altered the average intratumoural pH value that was almost equal to the value of the subcutaneous model (Figure 3C). As for the subcutaneous model, co-injection with GFP-AD-MSC was associated with the appearance of cells with fibroblast-like morphology within the tumour, forming septum-like structures (Figure 3D, arrows) that also caused a significant increase in tumour volume, as assessed by the luciferase assay (Appendix A).

Furthermore, as with subcutaneously injected xenografts, we indirectly demonstrated the activation of a marker of inflammation, upstream of the expression of IL6 and IL8 [45]: NF-kB. We specifically looked for its expression in tumour-associated cells with a fibroblast-like morphology, possibly GFP-AD-MSC in mice of group #4 (with GFP-AD-MSC), and in the same area of the tumour expressing the acid-related marker V-ATPase (Figure 3D). NF-kB activation was revealed by the detection of the intranuclear localisation of RelB (Figure 3D, black arrows in the magnified panel, at the bottom and on the right). This time, we used immunostaining since the mRNA extraction from intraosseous tumours in mice did not work, possibly due to the strong degradation activity that is required to extract the cells embedded in the bone matrix. Then, to investigate the preliminary step for the formation of lung metastases, we also compared the number of circulating tumour cells (CTC) in animal groups #3 (w/o GFP-AD-MSC) and #4 (with GFP-AD-MSC), by using the ParsortixTM microfluidic platform as previously described [40], and we found an increased trend in CTC number in group #4 (around 50 more than to group #3). Most importantly, group #4 exhibited a more rapid development of lung metastasis as detected by the quantification of the luciferase signal of luc-143B cells (Figure 3E) and immunostaining of human vimentin in the lung (Figure 3F). In the immunostained sections, we measured both the area covered by tumour nodules in the lung and the number of identified lesions (Figure 3G). In both cases, group #4 (co-injected with GFP-AD-MSC) showed significantly higher values with respect to group #3 (w/o GFP-AD-MSC). Furthermore, the group #4 showed a reduced overall survival (metastasis-free) with respect to group #3 (median survival 13.5 vs. 18 days, respectively, Figure 3H). 

### 3.4. Acid-Stimulated MSC Are Crucial for the First Step of the Metastatic Process

To further confirm and dissect the role of acid-activated MSC in the invasive, migratory, and metastatic potential of OS cells, we used 3D models mimicking the tumour mass in xenografts. We used acid-treated GFP-AD-MSC/143B mixed spheroids that appeared with a disperse morphology with respect to neutral conditions (Figure 4A), both at 0 and 3 days. Furthermore, at low pH, we observed a reduction in gap junction formation, as assessed by transmission electron microscopy and connexin-43 immunostaining (Figure 4B,C). These results preliminary suggest an increased ability of acid-treated cells to detach from the tumour mass. Then, to thoroughly evaluate tumour cell escape from the acid-secreting tumour mass, we used two different 3D microfluidic models. 

We first injected a population of mixed cells (luc-143B stained with DIL and GFP-AD-MSC) in Matrigel^®^, as a single-cell suspension, into a microfluidic chamber that was passively perfused with neutral or acidic medium (Figure 5A). 

We used pH 6.0, assuming that during the perfusion of the channel filled with Matrigel^®^ at pH 7.4, the acidic pH would have risen to around an average pH value, such as 6.7, thereby mimicking the perfusion of the adjacent acidic interstitial fluid, in the TME. Acidic perfusion significantly increased luc-143B cell migration (Figure 5B), even more than IL6 that was used as a positive control (Figure 5B). To further simulate 3D tumour growth and assess the potential of tumour cells to escape the primitive tumour mass and reach the circulatory system, we made use of an OrganoPlate Graft^®^: heterotypic spheroids were placed in the central grafting chamber (see Figure 5C for a schematic representation) and we assessed the number of cells that were able to invade the underlying gel, in the presence of acidic culture medium (pH 6.0) perfusing the Matrigel^®^ under the tumour mixed spheroids. We also did/did not add Tocilizumab, an IL6 neutralising antibody, to assess how much the acid-induced migration ability relied on the secretion of this chemokine. Under acidosis, tumour cells dramatically invaded the surrounding gel and Tocilizumab deeply and significantly impaired such behaviour (Figure 5D). 

Overall, these data suggest that tumour spheroids and 3D microfluidic models closely resemble the physiological TME, and that IL6 provides a therapeutic target to decrease the metastatic potential of cancer cells that survive acidosis. 

## 4. Discussion

We have previously demonstrated in vitro that highly proliferating and glycolytic OS cells acidify the surrounding microenvironment. In turn, extracellular acidosis can stimulate normal mesenchymal cells infiltrating the tumour to release mitogenic and chemotactic factors, via the activation of the NF-kB inflammatory pathways [24]. However, the role of acidosis in promoting lung metastasis via the reactive mesenchymal stroma has never been investigated, neither in complex in vitro 3D models, nor in in vivo models. Furthermore, with the exception of a few studies, the cytokines IL-6 and IL8, which are known negative prognostic markers for OS [37,46], have always been assumed to be produced by OS cells rather than by the activated mesenchymal stroma. In this context, it should be emphasised that the inclusion of the inflamed stroma in cancer modelling is crucial. Inflammation is a hallmark of cancer and is essential in cancer development and progression, even in those without obvious signs of inflammation and infection [47]. Among the sparse literature on the effects of MSC-derived IL6 on OS cells, the study of Tu et al. demonstrated the involvement of STAT3-mediated regulation of the expression of cyclin D, Bcl-xL and surviving [48]. However, in the cited study, the mechanism was investigated independently of the presence of an acidic microenvironment, which is instead important to better recapitulate the tumour context and to determine the effects of MSC exposed to excess protons: a decrease in local pH is per se an inflammatory stimulus independent of the presence of pre-existing bacterial endotoxins or pro-inflammatory cytokines, thereby causing the release of various enzymes during phagocytosis, the damage of vasculature and other surrounding tissues, and the prolonging of the healing process by stimulating new inflammatory reactions [49]. 

First, in a series of OS patients, we confirmed the correlation between high circulating levels of IL6 and IL8 and the tumour presence. Their decrease at T1 could be due to an indirect effect of chemotherapy, used in combination with surgery, rather than due to eradication of the tumour and of tumour-derived acidosis. However, at diagnosis, we found a significant correlation between their serum levels and FDG-PET signal, suggesting that their release is downstream of tumour glycolysis and lactic acid production, and thus, of tumour-derived acidosis. In the same series of patients, we also confirmed previous data on a correlation between circulating levels of another crucial inflammatory marker, the C-reactive protein, and reduced disease-free survival. Of note, previous authors also reported that the FDG-PET signal in OS was correlated with the risk of metastasis [50].

Inflammatory cytokines might not be the only mediators released by MSC in response to acidosis. In this context, EV have recently been considered as very important vehicles of mediators for cell–cell interactions in response to stress [51] and between MSC and OS cells [22], also with specific regard to the predisposition of the metastatic niche [22,52]. We then isolated EV from the serum of the same series of patients. Analysis of EV identified specific miRNAs that were strongly decreased after tumour removal. Among these, in the context of this study, mir136-5p is of particular interest because it has been previously associated with inflammatory processes. This miRNA was also found in EV derived from MSC cultured in vitro [53]. These data constitute a preliminary investigation in a field still unexplored but that could have significant implications both in the diagnosis and therapy of OS patients, laying the groundwork to identify new therapeutic targets to counteract metastasis occurrence. 

We then sought to confirm the relationship between inflammation, acidosis and tumour progression using animal models. First, we tested whether the metabolic activity of OS cells recreated an acidic microenvironment also in vivo. pH values varied greatly from area to area, but were quite reproducible among the xenografts within the same group, around 6.86. This value was not influenced by the site of tumour injection, nor by the presence of MSC, since it was very similar when subcutaneously injected with the 143B cells alone and in the orthotopic model injected with the mixed cell population, and is consistent with values previously measured by using needle microelectrodes in human soft tissue sarcomas (7.01 ± 0.21) [54], and in feline fibrosarcoma (6.14, min 6.08, max 6.84) [55]. 

We then added the GFP-AD-MSC to the subcutaneously inoculated cell population. In the following days, to mimic MSC arriving and infiltrating the tumour from the bloodstream in humans, we also injected GFP-AD-MSC into the tail vein [20,22]. In our model, injection of AD-MSC was associated with the appearance of cells with fibroblast-like morphology within the tumour that tended to align to form septa, likely to restrain tumour progression, as part of the protective response against neoplasia and malignancy [56]. Treatment with GFP-AD-MSC also significantly increased the tumour mass. However, this effect was not due to acidosis, because when we simulated the formation of a tumour mass in vitro with spheroids, GFP-AD-MSC increased the portion of Ki-67 positive tumour cells only at neutral pH. This result was not unexpected since, previously, we showed in 2D cultures that acidosis per se inhibits rather than induces the growth of cells of mesenchymal origin, also highly proliferating like sarcoma [13,57]. From this result, we can infer that the increase in mass observed in vivo is likely due to the secretome of MSC in areas with a pH closer to neutrality within the tumour. On the opposite side, in acidic areas, the growth of tumour mass is likely to be slowed. However, acidosis-induced secretome from MSC in the same areas does trigger the release of inflammatory mediators and NF-kB activation: in the subcutaneous model, we found a clear correlation between expression levels of V-ATPase, a protein largely involved in tumour acidifying activity that pumps excess protons out of the cell, and IL6 expression levels, indirectly suggesting that a large amount of the IL6 released into the TME is produced by MSC. 

In order to study the mechanism in a realistic setting, we then used an orthotopic model that forms spontaneous lung metastases [22]. First, once we obtained the tumour outgrowth in bone, in addition to confirming the formation of an intratumoural acidic pH, as in the subcutaneous tumour, in GFP-AD-MSC co-inoculated mice, we observed the appearance of fibroblast-like cells that arranged themselves within the tumour to form septa-like structures and a larger tumour volume with respect to the group injected only with tumour cells. In addition, fibroblast-like cells, adjacent to tumour areas that were highly positive for V-ATPase expression, also showed an activated NF-Kb inflammatory pathway, as demonstrated by the presence of a RelB-associated nuclear signal. NF-κB is an essential transcription factor for inflammatory responses, and is one of the most important molecules linking chronic inflammation to cancer [47], upstream of IL6 release. In this context, recent analyses of OS transcriptome have highlighted IL6 as one of the risk factors most frequently associated with the onset of metastasis [58,59], and this effect is consequent to the activation of a TET2-dependent mechanism [60]. In line with this observation, in our model, co-injection with GFP-AD-MSC was associated with a trend of increase in circulating OS cells and with clearly increased lung metastases, which were more numerous and larger, and negatively affected animal survival. 

By using OS spheroids already successfully employed to recapitulate OS stemness and assess acid-induced metabolic alterations [12,61], we demonstrated that acid-treated GFP-AD-MSC/143B heterotypic spheroids resulted in a more dispersed morphology and reduced the formation of gap junctions, suggesting a reduction in cell–cell adhesion, and thus, a greater tendency for tumour cells to leave the primary tumour site. Turning to another 3D model, based on microfluidics that can mimic perfusion and gradient formation in the TME, we have shown that acidosis significantly increases the local invasive capacity of tumour cells, both as single cells and when embedded in a 3D spheroidal tumour mass. The latter pattern is particularly relevant, as acidosis can result in increased invasiveness, especially at the edge of the tumour, where tumour cells express higher levels of proteins associated with elevated glycolytic metabolism and acid production [62]. Finally, by using the same 3D microfluidic cultures, we evaluated the efficacy of a therapy against one of the identified inflammatory cytokines to impair metastasis formation induced by the local tumour acidosis: anti-IL6 antibody completely impaired OS migration. This therapeutic approach is already under evaluation in clinical trials for several cancer types, but is still being considered for sarcomas. 

## 5. Conclusions

In this study, we have demonstrated that, in OS, the mesenchymal stroma within the TME reacts to tumour-derived acidosis by triggering an inflammatory response that, in turn, exacerbates tumour cell escape from the primary site and the formation of lung metastasis. In the future, the identification of subsets of patients with particularly glycolytic and acidic tumours, which could be detected by advanced imaging systems, could help in the development of personalised medicine approaches, defining protocols based also on targeted anti-inflammatory therapies, to complement conventional therapy. 

## Figures and Tables

**Figure 1 cancers-13-05855-f001:**
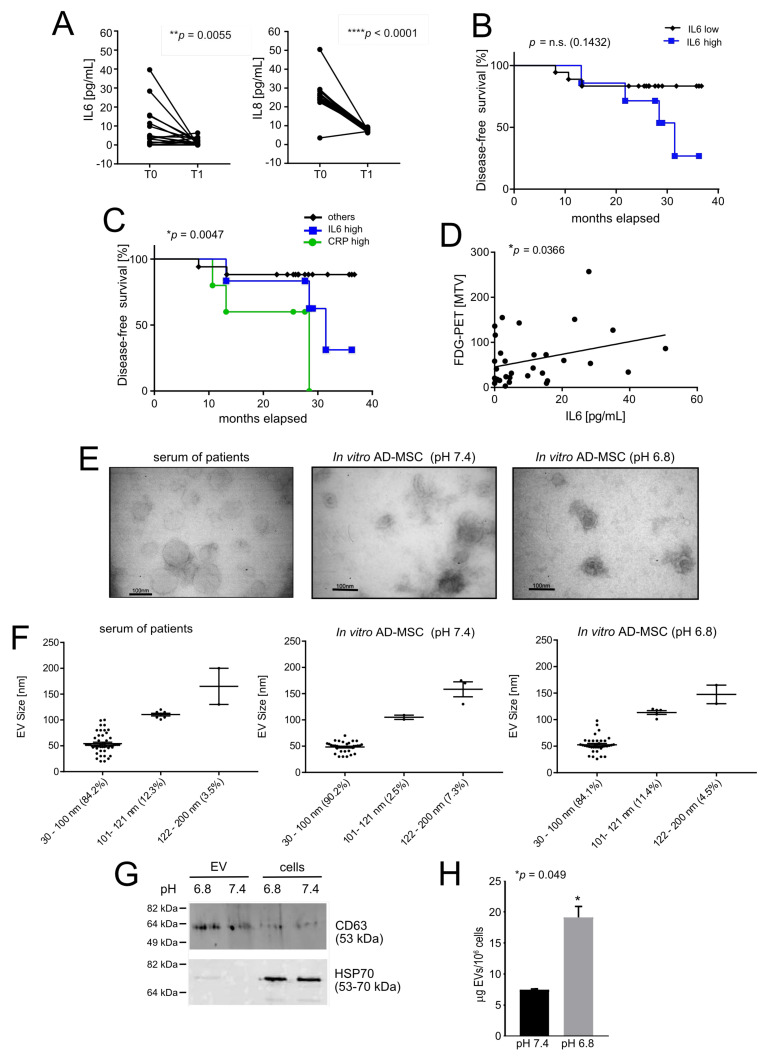
Systemic inflammatory mediators in OS patients that may be related to intratumoral acidosis. (**A**) ELISA of IL6 and IL8 in the serum of 18 OS non metastatic patients, at diagnosis (T0) and at the end of the therapeutic protocols, after tumour removal (T1) (Mann–Whitney U test, ** *p* > 0.01 and **** *p* < 0.0001); (**B**) Survival curve (disease-free) in OS patients that were metastasis-free at diagnosis and that showed high (≥4.3 pg/mL) or low (<4.3 pg/mL) circulating levels of IL6 (Kaplan–Meier curve, Log-rank Mantel–Cox test); (**C**) Survival curve (disease-free) in OS patients that were metastasis-free at diagnosis and that showed high circulating levels of IL6 (>4.3 pg/mL), or of C-reactive protein (CPR, >2 mg/dL) (Kaplan–Meier curve, Log-rank Mantel–Cox test, * *p* < 0.05); (**D**) Correlation between IL6 serum levels in OS patients and 18F-FDG PET/CT analysis, as measured by metabolic tumour volume (MTV) (Spearman Correlation test, * *p* < 0.05, *n* = 34 for IL6); (**E**) Representative images of transmission electron microscopy analysis of EV isolated from serum of OS patients and from AD-MSC cultured at different pH (scale bar 100 nm); (**F**) Percentage distribution of vesicles in different size classes; (**G**) Representative image of Western blot analysis of CD63 and HSP70 in EV isolated from AD-MSC and the relative cell lysates, cultured at different pH; (**H**) Total protein content in EV isolated from AD-MSC cultured at different pH conditions (mean ± SEM, Mann–Whitney U-test, * *p* < 0.05, *n* = 3).

**Figure 2 cancers-13-05855-f002:**
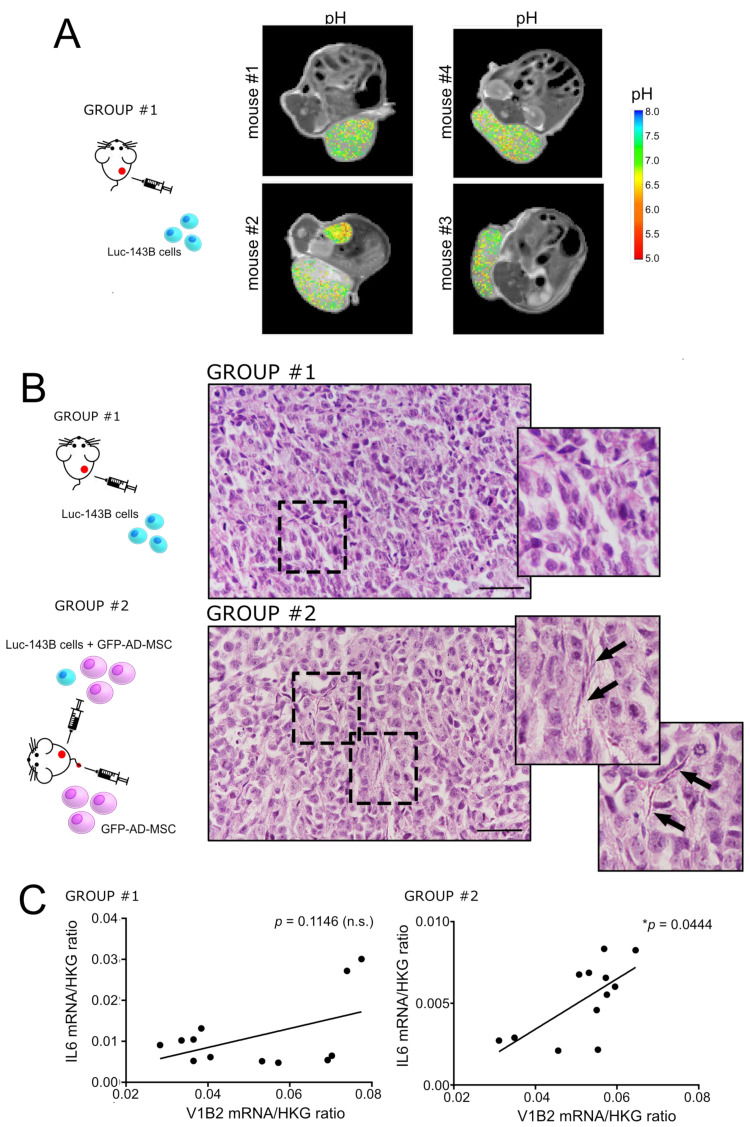
Intratumoural acidosis in subcutaneous OS promotes the expression of IL6 in tumour-associated MSC. (**A**) Representative tumour extracellular pH maps superimposed on anatomical T2w images for subcutaneous OS xenografts; in the same panel, the scheme of the experiment is also shown (trial n. 1, see Table 2); (**B**) Left side of the panel: representative scheme of the experimental animal trial n. 2 (see Table 2); right side of the panel: representative images of H&E staining of subcutaneous xenografts in animal trial 2, after the co-injection of AD-MSC and luc-143B OS cells (group #2) compared to xenografts obtained with homotypic cell population of tumour cells (group #1). Images with further magnification are shown in the right rectangle (arrows indicate cells with a fibroblast-like morphology); (**C**) Correlations of the mRNA expression levels of IL6 and V-ATPase V1B2 isoform in the xenografts of the two groups (Spearman correlation test, * *p* < 0.05, *n*= 12).

**Figure 3 cancers-13-05855-f003:**
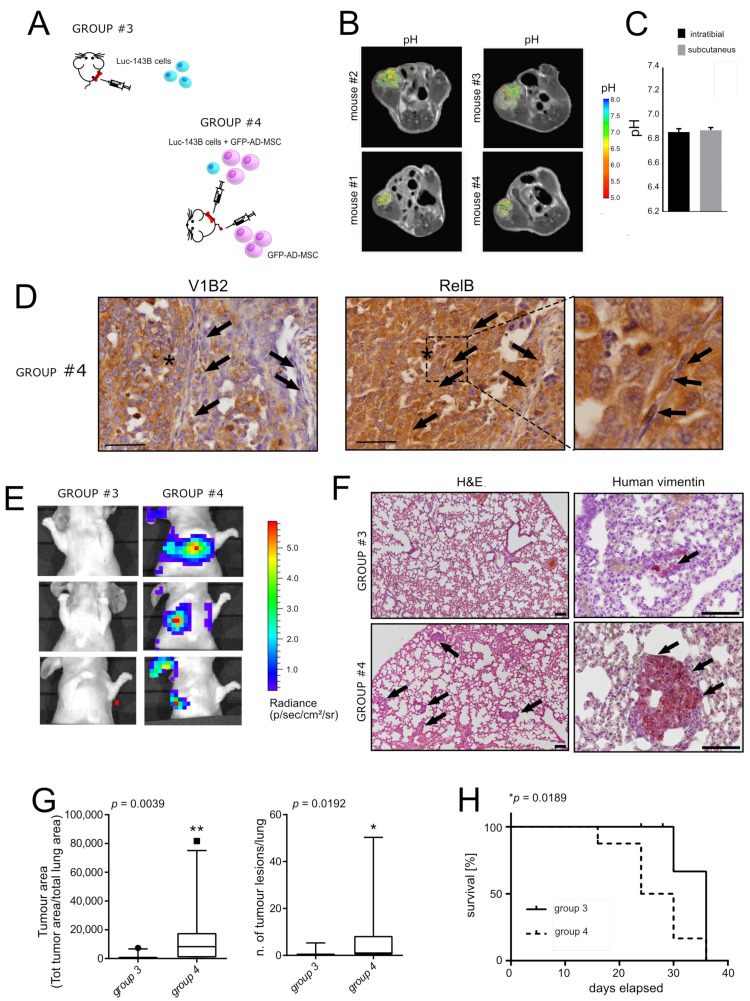
Intratumoural acidosis and co-injection with GFP-AD-MSC in intratibial OS promotes the expression of IL6 and fosters lung metastasis. (**A**) Representative scheme of the experimental animal trial n. 3 (see Table 1); (**B**) Representative tumour extracellular pH maps for intratibially injected OS xenografts, as measured by CEST-MRI; (**C**) Comparison of the intratumoural pH value of subcutaneously injected xenograft with luc-143B cells (group #1, see Table 1), and intratibially injected xenografts with luc-143B/GFP-AD-MSC (group #3, see Table 1), as measured by CEST-MRI (Mean ± SEM); (**D**) Representative images of the immunostaining of V1B2 V-ATPase and of RelB of intratibially injected xenograft after the co-injection of GFP-AD-MSC and luc-143B OS cells (group #3, see Table 1). RelB immunostaining image with further magnification is shown in the right rectangle (arrows indicate fibroblast-like cells, the asterisks indicate the area with possibly the low pH value since it is highly positive for V1B2 staining). Note the RelB nuclear localisation was found in cells with fibroblast-like morphology; (**E**) Images of the growth of lung metastases in three representative mice for each group of animal trial 4, with intratibially injected luc-143B cells (group #3), after 15 days, as monitored using bioluminescence imaging to which all the data shown are relative, and luc-143B cells co-injected with GFP-AD-MSC (group #4) (see Table 1); (**F**) Representative images of H&E staining (left panel) and of human vimentin immunostaining (right panels) of tissue sections of the lung (scale bar 100 µm, black arrows enlighten clusters of human tumour cells); (**G**) Graphs relative to the number of tumour lesions in the lung and to the tumour area with respect to the total area of the lung, as assessed by the detection of human vimentin signal and the quantification of positive spots and ROI area, respectively (*n* = 10 for group #3, and *n* = 11 for group #4, ** *p* < 0.01, Mann–Whitney U test, median and percentiles are shown); (**H**) Survival curve (lung disease-free). Uncensored events were considered when the analysed mice were positive to both luciferase assay and vimentin signal in the lung (Kaplan–Meier curve, Log-rank Mantel–Cox test, * *p* < 0.05).

**Figure 4 cancers-13-05855-f004:**
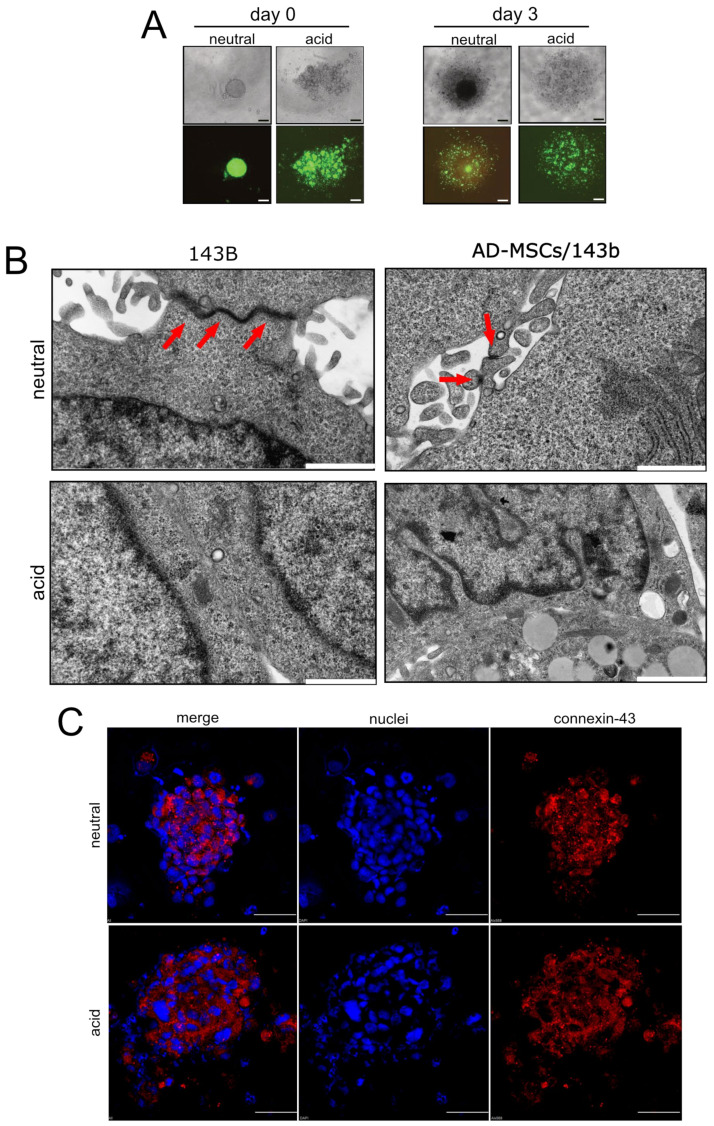
Extracellular acidosis favours the escape of tumour cells in mixed MSC/tumour spheroids. (**A**) Representative images of hanging-drop heterotypic spheroids (luc-143B + GFP-AD-MSC) cultured under neutral or unbuffered conditions (scale bar 50 µm); (**B**) Transmission electron microscopy images of homotypic spheroids (only 143B cells) or mixed spheroids (AD-MSC/143B cells) cultured at different pH conditions. Representative images. Gap junctions that were clearly visible at pH 7.4 completely disappeared at acidic pH (scale bar 2 µm); (**C**) Immunofluorescence staining of the gap junction protein connexin-43 in cytocentrifuged and fixed mixed spheroids (AD-MSC/143B cells). Representative images. The expression of connexin43 appeared reduced and more dispersed in acidic conditions with respect to neutral conditions (scale bare 50 µm).

**Figure 5 cancers-13-05855-f005:**
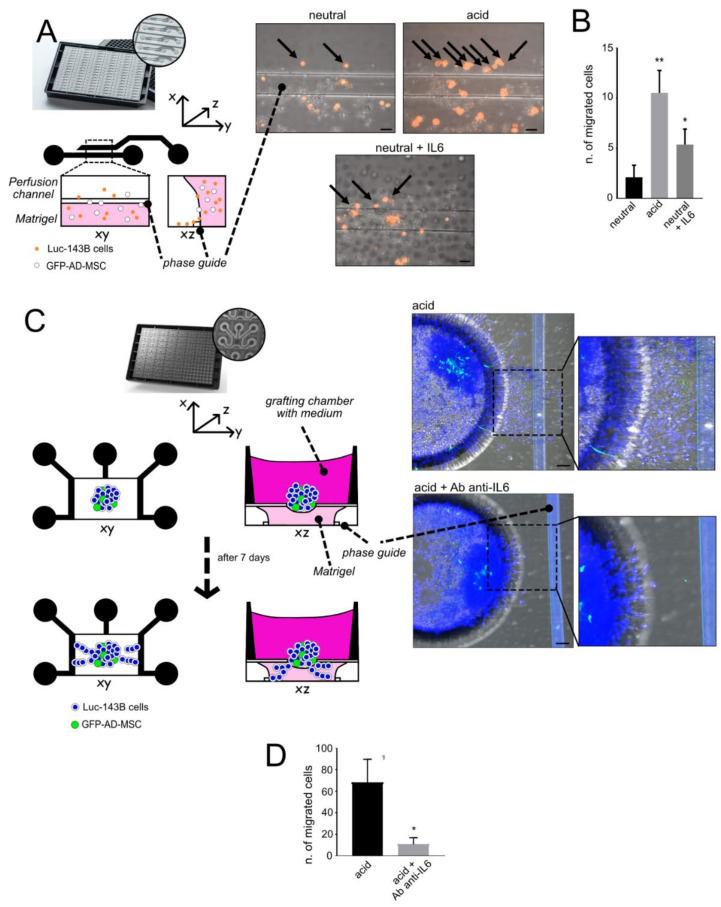
Extracellular acidosis favours the migration of tumour cells in 3D microfluidic cultures mimicking the acid TME. (**A**) Left panel: explanatory drawing of the experiment performed in the 2-lane microfluidic OrganoPlate^®^; right panel: representative images of the two channels in the microfluidic chamber in different conditions (black arrows indicate migrated tumour cells). Scale bar 20 µm; (**B**) Quantification of cells migrated in the perfusion channel of the experiment shown in (**A**), only cells in the perfusion channel were counted (mean ± SEM, ** *p* < 0.01 and * *p* < 0.05 vs. neutral, Mann–Whitney U test); (**C**) Left panel: explanatory drawing of the experiment performed in the OrganoPlate Graft^®^; right panel: representative confocal images (and magnifications) of the spheroids placed in the grafting chamber, after 7 days, in different conditions, and stained with Hoechst 33342. Scale bar 50 µm; (**D**) Quantification of the cells invading Matrigel^®^ in the grafting chamber, under the spheroids (* *p* < 0.05, Mann–Whitney U test).

**Table 1 cancers-13-05855-t001:** Clinical and pathological features of OS patients.

Variable	Total Series (*n* = 40)	Subseries Screened for miRNA Content in Circulating EV (*n* = 10)
Age (y)		
≤14 >14	21 19	4 6
Sex		
Male Female	24 16	5 5
Anatomic site		
Femur Tibia Humerus Pelvis Fibula Radius	24 10 2 2 1 1	6 3 0 1 0 0
Histological subtype		
Osteoblastic Chondroblastic Fibroblastic Osteoblastic/Chondroblastic Osteoblastic/Fibroblastic Osteoblastic/Telangiectatic	24 1 1 5 4 5	5 0 0 2 1 2

**Table 2 cancers-13-05855-t002:** List of xenograft experiments.

Animal Trial n.	Representative Scheme of the Experiment	Site of Injection and Cells (n)	N. of Groups (n. Mice/Group), Code Group	Endpoints (Days From Injection)	Type of Assay
1	Figure 2A	Subcutaneous, one-off (luc-143B 75,000)	1 (4) Group #1	21	MRI-CEST
2	Figure 2B	Subcutaneous, one-off (luc-143B 75,000/GFP-AD-MSC 225,000) + tail vain once/week (GFP-AD-MSC 225,000)	2(6) Group #1 and #2	22	RNA lysis from frozen tissues; Luc. Assay to the site of injection; E&E from paraffin-embedded tissues
3	Figure 3A	Intratibial, one-off (luc-143B 75,000/ GFP-ADMSC 225,000) + tail vain once/week (GFP-AD-MSC 135,000)	1 (4) Group #3	21	MRI-CEST (bone)
4	Figure 3A	Intratibial, one-off (luc-143B 75,000/GFP-AD-MSC 225,000) + tail vain once/week (GFP-AD-MSC 135,000)	2 (12 for group #3 and 13 for group #4) Group #3 and #4	8 (bone) 8-12-15-18 (lung)	Luc Assay to the site of injection and to the lung; H&E to the site of injection and to the lung; Immunohistochemistry for V1B2 and RelB at the site of injection, for vimentin at the lung

**Table 3 cancers-13-05855-t003:** Top most differentially expressed miRNAs between T1 and T0. SD, Standard deviation across the groups; fold changes between the two groups; *p*-value, *t*-test; adjusted *p*-value, Benjamini–Hochberg-adjusted (BH adj) test.

miRNA Name	Standard Deviation (SD)	Fold Change	*p*-Value (*t*-Test)	*p*-Value (BH adj)
** *Decreased miRNA (from T0 to T1* ** **)**				
hsa-miR-376a-3p	0.68	−2.4	0.00022	0.035
hsa-miR-154-5p	0.55	−2.8	0.00041	0.035
hsa-miR-136-5p	0.63	−2.0	0.00061	0.035
hsa-miR-376c-3p	0.81	−2.3	0.0010	0.043
hsa-miR-495-3p	1.2	−3.0	0.0047	0.14
hsa-miR-320c	0.27	−1.2	0.0048	0.14
hsa-miR-99b-5p	0.65	−1.6	0.0080	0.16
hsa-miR-409-3p	0.95	−2.3	0.0095	0.16
hsa-miR-485-3p	0.61	−2.1	0.016	0.21
hsa-miR-199a-3p	0.34	−1.2	0.031	0.30
hsa-miR-186-5p	0.52	−1.3	0.031	0.30
** *Increased miRNA (from T0 to T1)* **				
hsa-miR-144-5p	0.94	1.9	0.015	0.21
hsa-miR-30a-5p	0.51	1.4	0.010	0.16
hsa-miR-22-3p	0.28	1.2	0.017	0.21
hsa-miR-374a-5p	0.23	1.1	0.020	0.23
hsa-miR-20b-5p	1.3	2.3	0.027	0.29
hsa-miR-139-5p	0.50	1.4	0.0096	0.16
hsa-miR-18b-5p	0.41	1.2	0.035	0.31
hsa-miR-126-5p	0.31	1.2	0.036	0.31
hsa-miR-424-5p	0.46	1.4	0.0064	0.16

## Data Availability

The data presented in this study are available from the corresponding author on reasonable request. Part of the datasets generated and/or analysed during the current study are available in the ‘Figshare’ repository (https://figshare.com/s/a2a31b153d50bde31e4e, Digital Object Identifier 10.6084/m9.figshare.16593020) (17 October 2021).

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
