# Peer review of "The Release of Inflammatory Mediators from Acid-Stimulated Mesenchymal Stromal Cells Favours Tumour Invasiveness and Metastasis in Osteosarcoma"

_cancers, 2021, doi:10.3390/cancers13225855_

Round 1
Reviewer 1 Report
General Comments: In this manuscript, Avnet et al. pose that acid-stimulated mesenchymal stromal cells release inflammatory mediators that promote tumor invasiveness and metastasis in human osteosarcoma (OS). They make this association first in a patients series from 2015-2018 by observing that high serum IL-6 levels correlated with disease free survival and FDG-PET avidity, an indirect measure of total lesion glycolysis (TLG) and acidosis. Next the study the presence of an acidic environment with OS xenografts (SQ and orthotopic) using MRI-CEST pH imaging techniques and correlate this data to tumor glycolysis, acidosis, and inflammatory markers. Finally, they examine the effects of anti-IL6 therapies on cell migration using 3D microfluidic models that harbor MSC/OS cell mixtures. Overall, this is a well written manuscript that applies a number of interesting models to better understand how an acidic TME affects OS cell metastasis. I do have some concerns with Figure 1 that seems to overestimate the link between FDG-avidity and the levels of IL6 and IL8, and this dampens my enthusiasm for this manuscript somewhat. Also, the tumor heterogeneity of MRI-CEST data in Figure 2 presents some challenges. Last, while I commend the authors for studying OS cells (and MSCs) using 3D models (e.g., spheroids and microfluidics), a number of confounding variables, such as hypoxia, may also be contributing the pro-migrating effects. Though perhaps the subject of future studies, it would have been interesting to merge their MRI-CEST data onto a spatial image-omics (SIO) analysis to explore whether local regions of relative intra-tumoral acidosis correlate with varied transcriptomic patterns in the co-localized regions. Major Comments and Concerns: 1. Please explain the high ratio of MSCs to OS cells used for the in vivo animal studies. I suspect the number of MSCs used is far higher that what is observed in human OS. 2. Figure 1, panel B: Why not show the KM curve of IL6-high compared to IL6-low, which would seem to be the most appropriate comparison? 3. Figure 1, panel C: The positive correlations appear to be driven by a single outlier. Ignoring that patient, there doesn’t appear to be a statistically significant correlation between FDG-PET and IL6 or IL8, respectively. 4. Explain why adipose-derived MSCs were used instead of bone marrow MSCs, since there might subtle be phenotypic differences that affect tumor behavior. 5. In addition to imaging-based measures of intratumoral acidosis, direct measures from the animal tumor tissues might have been helpful. Indirect measures, such as V-ATPase pump V1B2 isoform are relatively unconvincing. 6. The degree of acidosis observed in the spheroids was higher than what was observed in the animal tumors, indicating they might not adequately resemble the intratumoral pH. Minor Comments: 1. Table 2: Recommend using English terminology instead of “una tantum”.

Author Response
> We thank the reviewer for the appreciation of the manuscript. We further reviewed the correlation of FDG-avidity and IL6 and IL8 levels by performing a statistical analysis of outliers. After exclusion of outliers, it was indeed found that IL8 values do not correlate with FDG-PET values. However, IL6 still showed a significant correlation with glucose uptake (by Log-rank Mantel-Cox test). Regarding the other comments, we consider them as valuable insights for further study.
Major Comments and Concerns:
- Please explain the high ratio of MSCs to OS cells used for the in vivo animal studies. I suspect the number of MSCs used is far higher that what is observed in human OS.
> We used such a high ratio of MSC to tumor cells because 143b cells have a much higher proliferation rate and, within days, this ratio likely reverses completely. We also documented this inverted ratio over time by performing flow cytometric analysis of cells isolated from spheroids: after 14 days of culture under standard conditions, GFP-MSC isolated from MSC/143b mixed spheroids (seeded 3:1) accounted for 10% of the total cell population, whereas 143b cells accounted for 90% of the total cell population. These data have been included in another article, in preparation.
> In the text of the manuscript, at page 4, in the method section on spheroids (§2.4), we added the following sentence (the new text is in UPPERCASE): “Homotypic luc-143B spheroids were obtained seeding 5 x 103 cells/well, whereas heterotypic luc-143B/GFP-AD-MSC were seeded in a 1:3 ratio (5x103 tumour cells + 1.5 x 104 stromal cells). WE USED SUCH A HIGH RATIO OF MSC TO TUMOR CELLS BECAUSE 143B CELLS HAVE A MUCH HIGHER PROLIFERATION RATE AND, WITHIN DAYS, THIS RATIO LIKELY REVERSES COMPLETELY.”
> In the text of the manuscript, at page 7, in the method section on in vivo models (§2.11), we added the following sentence (the new text is in UPPERCASE): “For the in vivo models we used both subcutaneous and intratibial injections of homo-typic cell population or of heterotypic tumour/stromal mixed cell populations (Table 2). AS FOR THE SPHEROIDS, WE USED SUCH A HIGH RATIO OF MSC TO TUMOR CELLS BECAUSE 143B CELLS HAVE A MUCH HIGHER PROLIFERATION RATE AND, WITHIN DAYS, THIS RATIO LIKELY REVERSES COM-PLETELY.”
- Figure 1, panel B: Why not show the KM curve of IL6-high compared to IL6-low, which would seem to be the most appropriate comparison?
> As suggested by the reviewer, we also added the Kaplan Mayer curve of IL6-high compared to IL6-low in Figure 1B and described the relative results in the revised version of manuscript at page 12: “Although not statistically significant, we observed a trend of decrease for the percentage of disease-free survival in the IL6-high group of patients in respect to the IL6-low group (Fig. 1B)”.
- Figure 1, panel C: The positive correlations appear to be driven by a single outlier. Ignoring that patient, there doesn’t appear to be a statistically significant correlation between FDG-PET and IL6 or IL8, respectively.
> As suggested by the reviewer, we calculated outliers by using the Grubbs' test (alpha = 0.05) for all the data sets. The results revealed a single outlier for both IL6 and IL8 correlation with FDG-PET. After the exclusion of this outlier, serum levels of IL6 significantly correlated with FDG-PET (p = 0.0366) whereas we did not find a significant direct correlation for IL-8. We changed the text accordingly at page 12.
- Explain why adipose-derived MSCs were used instead of bone marrow MSCs, since there might subtle be phenotypic differences that affect tumor behavior.
> We obtained GFP-AD-MSC thanks to the collaboration with Dominici et al. who already published their use in previuos studies on sarcomas (Guiho et al, Int J Cancer 2016;139:2802-11; Grisendi et al, Cancer Res 2010;70:3718-29; Grisendi et al, Stem Cells 2015;33:859-69). Furthermore, the use of AD-MSC for the study on the paracrine interactions with sarcoma cells is an already well established model (Baglio et al. Clin Cancer Res 2017;23:3721-3733; Mannerstrom et al. Epigenetics 2019;14:352-364; Wang wt al. Fornt Cell Dev Biol 2020;8:353; Wang et al. Oncotarget 2017;8:23803-23816).
- In addition to imaging-based measures of intratumoral acidosis, direct measures from the animal tumor tissues might have been helpful. Indirect measures, such as V-ATPase pump V1B2 isoform are relatively unconvincing.
> We agree with the reviewer. However, direct live measurements of pH, with macro or microelectrode, would imply opening the tumour resulting in possible confounding factors that may affect pH value measurement, like the exposure to CO2 in the air that greatly impacts on pH stability, and the bleeding (bicarbonate content in the blood). We have already experienced this type of technical issues by measuring pH by electrode in cat fibrosarcoma (Martano et al Biomed Res Int 2019;8275938). Additionally, pH values measured by electrode are negatively affectd by the mixed contribution of the intracellular and the extracellular compartments that is not possible to disentangle. Adversely, live imaging is a noninvasive method that provide accurate and reliable pH measurements of only the extracelllular tumor compartment and we believe it is, therefore, the best for our model. And we considered V-ATPase data only as a confirmation of the data obtained with live imaging.
- The degree of acidosis observed in the spheroids was higher than what was observed in the animal tumors, indicating they might not adequately resemble the intratumoral pH.
> We measure the pH of the supernatant of spheroids at the end of the culture, which corresponded to 6.6 and 6.7. However, we have never measured the luminal/intratumoral pH of spheroids and do not know how much it is, as it could be slightly higher or slightly lower than that of the unbuffered supernatant. Imaging analysis of xenografts revealed a pH around 6.88 ± 0.04, meaning that the minimum estimated value for the average pH is 6.84. However, intratumoral pH can also change considerably between different areas of the tumour. Therefore, we believe that a difference of 0.1-0.2 (between the average intraspheroid pH and the average intraxenograft pH) very likely falls within this range.
- Minor Comments: 1. Table 2: Recommend using English terminology instead of “una tantum”.
> The text was modified as suggested.
Reviewer 2 Report
The presented article is original and has highs scientific soundness, and the study was perfectly designed. The tested hypothesis was related to the influence of tumour glycolysis/acidosis on the metastases. The Authors used biomarkers of inflammation and tested them in liquid biopsies. They confirmed that elevated levels of both C-reactive protein (CRP) and IL6, not IL-8, correlate with an increased risk of lung metastasis. Moreover, the authors indicated that miR-136-5p could be a significant factor with diagnostic and prognostic features that regulate the expression of pro-inflammatry cytokines (including IL-6). This work emphasizes the development of more efficient protocols of OS treatment, especially in terms of targeted anti-inflammatory therapies.
Minor improvements can be made within The Materials And Methods section:
Please add the information regarding sensitivity of immunoenzymatic assays used for the detection of IL-6 and IL-8.
Please describe the primers used for the detection of transcripts (mRNA level) more precisly: add a reference to GeneBank, information regarding primers loci and amplicon length.
Add information about cat. no. of antibodies used for western blot analysis.
Additionally, please improve some data presentation:
Fig. 1. D: Could you please add information regarding EVs size?
Fig.1.E: The western blot for CD63 is poor, and bands are not homogenous with a high background. Could you improve/repeat this measurement? Moreover, provide information about the molecular weight of the bands detected – please include the molecular mass ladder.
Fig. 3D. The scale bar on microphotographs from H&E staining is barely seen.
Author Response
> We thank the reviewer for the appreciation of the manuscript.
Minor improvements can be made within The Materials And Methods section:
- Please add the information regarding sensitivity of immunoenzymatic assays used for the detection of IL-6 and IL-8.
> The sensitivities of immunoenzymatic assays to detect IL-6 and IL-8 are 0.7 pg/mL and 7.5 pg/mL, respectively. We add this information in the revised version of manuscript at page 4.
- Please describe the primers used for the detection of transcripts (mRNA level) more precisly: add a reference to GeneBank, information regarding primers loci and amplicon length.
> As requested by the reviewer, we added a new table in the Supplementary Materials containing all the requested information on primers.
- Add information about cat. no. of antibodies used for western blot analysis.
> According to the reviewer suggestions, we added in §2.10 methods section the information about cat. No. of the antibodies.
- Additionally, please improve some data presentation: Fig. 1. D: Could you please add information regarding EVs size?
> According to the reviewer suggestions, we added EVs size description in Fig. 1F.
- Fig.1.E: The western blot for CD63 is poor, and bands are not homogenous with a high background. Could you improve/repeat this measurement? Moreover, provide information about the molecular weight of the bands detected – please include the molecular mass ladder.
>According to the reviewer suggestions, we added the molecular weight of the proteins (CD63 ~53 kDa and HSP70 53~70 kDa) in Fig 1G. Furthermore, for the sake of clarity, the whole uncropped blot has been uploaded in the repositery data weblink, as mentioned at the end of the manuscript.
- Fig. 3D. The scale bar on microphotographs from H&E staining is barely seen.
> We adjusted the size of the scale bars as requested.
Reviewer 3 Report
In this paper, the authors revealed that acidic tumor microenvironment promoted invasion of human osteosarcoma cells via humoral factors and EV secreted by MSC infiltrating the tumor. This paper is very interesting, however, following comments need to be improved by the authors:
major
1) Is there any data showing that IL-6 whose expression was increased in the animal experiment shown in Fig. 2 is derived from GFP-AD-MSC? Is there data that MSC induces IL-6 and IL-8 expression in an acidic environment?
2) Have the authors confirmed that the "fibroblast-like morphology cells" observed in Fig.2B Group # 2 were GFP-AD-MSC? Are those cells GFP-positive?
3) In Fig.5B, it seems that the IL-8 or miR-136-5p-containing EV addition group or adding the supernatant of MSC cultured in an acidic environment to 143B cells are necessary.
minor
1) Fig.1D shows an electron micrograph observing the particle size of EV. Is it possible to analyze with a nanotracking system?
2) The WB data of CD63 in Fig.1E is unclear.
3) Table 3 should be easier to understand if it is divided into increased miRNA and decreased miRNA. Have the authors verified the changes in all 20 miRNAs with qPCR?
Author Response
> We thank the reviewer for the appreciation of the manuscript.
major
1) Is there any data showing that IL-6 whose expression was increased in the animal experiment shown in Fig. 2 is derived from GFP-AD-MSC? Is there data that MSC induces IL-6 and IL-8 expression in an acidic environment?
> We tried to stain GFP in the mice tissue section with several different types of antibody with no success. Thus, we do not have any direct method to distinguish from which cell type the detected mRNA of IL6 in xenografts is coming. However, the stronger and significant correlation of IL6 mRNA in the mice group with the co-injection with MSC in respect to the group of mice injected with the solely tumor cells suggests that MSC are the cells that most secrete this factor. Furthermore, in vitro, we already demonstrated and published that acid-stimulated MSC express (mRNA) and secrete significantly higher level of IL 6 than osteosarcoma cells (Avnet et al. Int J Cancer 2017;140:1331-1345). Finally, it is not possible to isolate MSC from the tumour tissue and extract the mRNA as a second step, because MSC share the same antigens of tumour cells since they have the same mesenchymal origin.
2) Have the authors confirmed that the "fibroblast-like morphology cells" observed in Fig.2B Group # 2 were GFP-AD-MSC? Are those cells GFP-positive?
> See the answer above.
3) In Fig.5B, it seems that the IL-8 or miR-136-5p-containing EV addition group or adding the supernatant of MSC cultured in an acidic environment to 143B cells are necessary.
> In this experiment we directly added MSC to the microfluidic cultures. Thus, IL8 or IL6 or miR-136-5p-containing EV are directly secreted from MSC included in the system, and exposed to acidosis. The condition with the solely IL6 at neutral pH have been added as a positive control for cancer cell migration.
minor
1) Fig.1D shows an electron micrograph observing the particle size of EV. Is it possible to analyze with a nanotracking system?
> According to the reviewer suggestions, we described EVs size in Fig 1F.
2) The WB data of CD63 in Fig.1E is unclear.
> For the sake of clarity, the whole uncropped blot has been uploaded in the repositery data weblink.
3) Table 3 should be easier to understand if it is divided into increased miRNA and decreased miRNA. Have the authors verified the changes in all 20 miRNAs with qPCR?
> According to the reviewer suggestions, we modified Table 3 to better highlight increased and decreased miRNAs.
> The analysis of miRNAs expression was performed and verified by qPCR (QIAGEN Genomic Services).
Round 2
Reviewer 1 Report
Accept the reviewer's response and recommend manuscript approval.
Reviewer 3 Report
I confirmed that the authors revised the manuscript appropriately.